# Cattle and Pigs Are Easy to Move and Handle Will Have Less Preslaughter Stress

**DOI:** 10.3390/foods10112583

**Published:** 2021-10-26

**Authors:** Temple Grandin

**Affiliations:** Department of Animal Science, Colorado State University, Fort Collins, CO 80523, USA; cheryl.miller@colostate.edu

**Keywords:** cattle, pigs, meat quality, handling, lameness, slaughter

## Abstract

Previous research has clearly shown that short-term stresses during the last few minutes before stunning can result in Pale Soft Exudative (PSE) pork in pigs or increased toughness in beef. Electric prods and other aversive handling methods during the last five minutes are associated with poorer meat quality. Handlers are more likely to use aversive methods if livestock constantly stop and are difficult to move into the stun box. Factors both inside and outside the slaughter plant contribute to handling problems. Some in-plant factors are lighting, shadows, seeing motion up ahead, or air movement. Non-slip flooring is also very important for low-stress handling. During the last ten years, there have been increasing problems with on-farm factors that may make animals more difficult to move at the abattoir. Cattle or pigs that are lame or stiff will be more difficult to move and handle. Some of the factors associated with lame cattle are either poor design or lack of adequate bedding in dairy cubicles (free stalls) and housing beef cattle for long periods on concrete floors. Poor leg conformation in both cattle and pigs may also be associated with animals that are reluctant to move. Indiscriminate breeding selection for meat production traits may be related to some of the leg conformation problems. Other on-farm factors that may contribute to handling problems at the abattoir are high doses of beta-agonists or cattle and pigs that have had little contact with people.

## 1. Introduction

Many research studies have previously shown that short-term stresses during the last few minutes before stunning may result in both poorer meat quality and severely compromised animal welfare. In pigs, multiple shocks from electric prods within five minutes before stunning resulted in higher lactate levels and more Pale Soft Exudative meat (PSE) [1,2,3]. Shocking pigs multiple times with an electric prod greatly increased lactate and glucose levels compared to low-stress handling and no use of electric prods. In cattle, short-term stresses shortly before stunning such as the use of electric prods or agitated behavior were associated with increased toughness in the meat [4,5]. The purpose of this commentary is to discuss the author’s observations of the increase in on-farm factors that may make animals more difficult to move at the abattoir.

When cattle or pigs are difficult to move and constantly keep stopping, handlers are more likely to use aversive methods to drive them such as electric prods or tail twisting [6]. Pigs that move easily also require fewer touches, slaps, or pushes. In this article, the author discusses some of the factors that are associated with animals that are difficult to move at the slaughter plant. Correcting these problems will help improve both welfare and meat quality.

There are factors both inside and outside the abattoir that can have an effect on the ease of animal movement. Major factors inside the plant are distractions such as sharp shadows on the floor, reflections on shiny metal, or a noisy vehicle near the lairage (stockyards) that may cause animals to stop [6,7,8,9]. Illuminating a dark restrainer entrance facilitated the movement of cattle and it reduced vocalization associated with electric prod use [7]. Other factors that can slow down animal movement are air blowing out through the stun box entrance towards approaching cattle or layout mistakes in the design of facilities [9]. Abattoir yards and lairages should also have non-slip flooring in pens, alleys, and stun boxes. If animals slip and fall on a slick floor, they are more likely to become stressed. It is also essential to have well-trained stock people who understand and use behavioral principles of moving livestock [9,10]. Training of stock people will improve livestock handling and reduce aversive methods of driving livestock [11].

The main emphasis of this article is to discuss on-farm factors such as housing problems, growth promotants, or overselection for production traits that may be associated with lame cattle and pigs that are reluctant to move. It contains both scientific studies and observations from the author’s experiences with handling livestock. Since the early 1970s, the author has consulted on improving livestock handing in abattoirs in the U.S., Europe, South America, Australia, and many other countries. High preslaughter standards for animal welfare are difficult to maintain if animals are lame, stiff, or have reduced mobility. These problems must be corrected at the farm of origin. The author has observed that handling problems at the abattoir are increasingly associated with breeding, feeding, or housing practices on the farm.

## 2. On-Farm Factors Associated with Handling Problems at the Abattoir

Within the last ten years, the author has observed that problems with cattle and pigs that are less willing to move have increased. Recently, a lairage manager at a large beef abattoir told the author that cattle from certain feedlots would immediately lie down after arrival. They were so reluctant to move that he had to get them up a few minutes before they went to the stunner. There are a variety of conditions that may have contributed to these handling problems. Over the years, the author has observed that these handling problems have slowly become worse. Issues with poor mobility of pigs and cattle have increased slowly and people did not notice it. The author calls this “bad becoming normal” [12].

In the U.S., the percentage of lame grain-fed cattle has increased. In 2020, only 74.5% of grain-fed beef cattle were free of lameness [13]. These data were collected during the months of July to October on 16,262 fed feedlot cattle that arrived at a large abattoir located in the Central Plains of the U.S. This area is in the heart of the U.S. feedlot industry. In the previous years of 2016–2019, 96.19% to 89.32% of the fed feedlot cattle that arrived at an abattoir were free of lameness [13]. There are also some dairies with high percentages of lame cows. Lame livestock that are reluctant to move may be more likely to have stressful aversive handling methods used on them at the slaughter plant. A recent Brazilian survey of 50 dairies showed that 41% of the cows were lame [14]. The percentage of lame cows varied from 13.8% in the best dairy to 64.5% in the worst dairy [14]. Recent studies conducted in the UK and Canada indicated that 31.8% of the dairy cows in the UK were lame [15] and 21% of the dairy cows kept in cubicles were lame [16]. Research also clearly shows that dairy producers will often greatly underestimate the percentage of lame dairy cows [17]. When lameness is actually measured, they will discover that the percentage of lame dairy cows may be double their estimate. 

### 2.1. Poor Structural Leg Confirmation

The author has observed that grain-fed market cattle or pigs, which are indiscriminately bred for growth, are more likely to be lame and reluctant to move. At one large abattoir, 50% of the incoming market weight pigs were lame. Approximately half of the pigs had poor leg conformation and exhibited traits such as legs too straight (post legged), collapsed ankles, or the feet were rotated. The problem probably starts with the sow herd. Breeding stock with poor leg conformation had a higher rate of culling due to lameness [18]. Pork and beef producer organizations have now recognized the problem and they have distributed leg conformation charts for producers to use when they select breeding stock [19,20]. The American Angus Association has an EPD for leg conformation [21]. It was created in response to producer reports that leg conformation was worsening in Angus cattle selected for rapid weight gain and large muscles. In Thailand, the author observed severe lameness issues in pigs that had been selected to have small feet.

### 2.2. Deficiencies in Housing Associated with Lameness or Swollen Joints

Poorly designed housing or lack of management of housing is associated with injuries to legs that may lead to lameness in both dairy cows and beef cattle. Housing fattening beef cattle on concrete for long periods of time can lead to swollen joints [22]. In dairy cows, free stalls (cubicles) that are either too small or poorly bedded are associated with more leg problems and swollen joints [23,24]. Farms with better bedding management had fewer cows with swollen joints [23]. Other factors that were associated with increased lameness were slippery floors and poor body condition [24]. In one survey, 40% of the skinny dairy cows with a body condition score under 2.5 were lame [24]. Improvements in flooring and maintaining cow body condition may also help reduce lameness. This shows the importance of good management on the farm for reducing lameness. Charolais bulls housed on a concrete slatted floor had significantly more lameness than bulls housed on deep litter [25]. The author has observed that lameness in cattle housed on concrete can be reduced by shortening the period of time they are kept on concrete. Covering the slats with rubber mats may also help reduce lameness. More research is needed to determine guidelines for the maximum length of time that fattening cattle should be housed on concrete floors.

### 2.3. Excessive Use of Growth Promotants and Handling Problems

Research has clearly shown that pigs fed high doses of beta-agonists are more likely to become fatigued and non-ambulatory [26]. Non-ambulatory pigs increase labor requirements at the abattoir and their welfare is severely compromised. Ractopamine and zilpaterol are feed additives that are used to increase the amount of muscle [27]. High doses of ractopamine were associated with greater difficulty in handling pigs [28,29]. Hot weather is also more likely to increase death losses in cattle fed ractopamine [30,31]. Handling problems observed by both the author and reports in the scientific literature both indicate that high doses, combined with hot weather over 32 °C, caused the most problems [8,29]. Pigs fed beta-agonists must be handled in a low-stress manner to prevent downed non-ambulatory animals [32]. Researchers at a Colorado feedlot also reported that feeding zilpaterol predisposed grain-fed cattle to heart problems [33]. Behavioral observations of cattle indicate that they may also have muscle stiffness. Feeding zilpaterol at the recommended label dose to grain-fed cattle resulted in 31% of the animals lying in an abnormal posture [34]. It is likely that the abnormal lying posture is related to attempts to reduce muscular discomfort. In one case, a group of cattle fed a high carbohydrate potato by-product diet combined with high doses of zilpaterol presented some cases where the outer hoof sloughed off [35,36]. Some specialists who are concerned about both meat quality and animal welfare may ask if transportation practices contribute to this problem. It is likely that transportation is not the main cause of this problem. Both the location of the abattoirs and the feedlots that supply the cattle had not changed. Both before and after the appearance of the handling problems, the cattle were transported the same distances from most of the same feedlots. The use of beta-agonists is banned in Europe and China [37]. They are legal in the U.S., Canada, Brazil, and many other countries [37]. It is important for readers in countries where beta-agonists are legal to be aware of possible handling and welfare problems. These problems are more likely to occur when higher doses are used.

## 3. The Concept of Biological System Overload

The author has been in the livestock industry for many years. In the 1970s through the 1990s, most welfare and handling problems in an abattoir were due to either poorly designed facilities, lack of equipment maintenance, or rough abusive handling by people [9]. Today, in a well-managed U.S. slaughter plant, handling problems with grain-fed cattle or pigs are more likely to be associated with on-farm factors. The problem may also be due to pushing the livestock to gain weight fast. This is accomplished by both genetic selection for production traits and feeding practices. The animal’s biology is pushed to the point where it starts to break down. Cardiac problems in cattle used to occur only at high altitudes. Researchers have found that they are now occurring at lower altitudes [38]. Heart problems in cattle associated with high altitude are heritable [39,40]. It is possible that heart problems are related to a greater emphasis on breeding cattle for large amounts of muscle mass. In 2015, veterinarians described a condition in cattle called fatigued cattle syndrome [36]. This is similar to problems with weak fatigued pigs. There are four factors that may have led to the relatively recent observations of more problems with both cattle and pigs that are reluctant to move:(1)Cattle fed to heavier weights at a younger age and more cattle fattened for highly marbled USDA prime beef [41,42];(2)Indiscriminate breeding and selection for growth and muscle mass in both species [42];(3)Feeding high-grain diets to cattle and a lack of roughages [25];(4)High doses of beta-agonists fed to both cattle and pigs [28,29].

It is the author’s opinion that pushing the animal’s biology until it starts to break down may be one of the most serious animal welfare problems [43]. These problems will also contribute to meat quality problems. The author has tracked non-ambulatory pigs through to the meat cutting floor and they had high levels of PSE. Increasing muscle growth with beta-agonists also resulted in increased beef toughness [44]. Producers should strive for optimum performance and not maximum growth and muscle. Producing animals that convert feed more efficiently into muscle is good from a sustainability standpoint because they eat less feed. There is a point where it is both not sustainable and bad for animal welfare. An animal that dies shortly before it is time to slaughter it wastes all the feed it has eaten.

## 4. On-Farm Behavioral and Management Factors

The discussions in the previous sections of this paper covered physical problems that made animals more difficult to handle. In this section, factors that are purely behavioral will be covered. The author has observed that pigs and cattle will move more easily during handling at the abattoir if they have become accustomed to people walking through them. An animal’s experiences on the farm will affect its behavior during handling in the future. When people regularly walk through the finishing pens several times each week, pigs will move more easily at the slaughter plant. Finishing-market-weight pigs that have been moved several times on the farm will be easier to move and drive through alleys in the future [45,46,47,48]. Pigs differentiate between a person walking in the aisle and a person walking through their pens. From the author’s experience on farms, pigs will be easier to handle if they become accustomed to quietly moving away when a person walks through their pens. The author has observed that cattle that have been extensively raised are sometimes difficult and dangerous to handle at the slaughter plant. Cattle can tell the difference between a person walking on the ground and a person riding a horse. Extensively raised cattle that have been handled with horses may have a greatly increased flight zone when they first encounter a person walking on the ground. The horse and rider were perceived as familiar and safe, and the person walking is new and novel [49]. Handling at the abattoir will be safer and cattle will be less stressed if they become accustomed to moving in and out of pens by people on foot before they arrive at a slaughter plant.

## 5. Two Observational Case Histories Where On-Farm Practices Had Significant Effects on Ease of Handling

The author consulted with a large pork plant that had severe problems with downed non-ambulatory finisher pigs and pigs that were difficult to move. To handle all the fatigued non-ambulatory pigs required five or six full-time people to stun the pigs in the lairage and transport the stunned animals to the bleeding area. After three changes were made on the farm, the numbers of downed non-ambulatory pigs dropped to the point where only one half-time person was required to handle downers. The three things the farms changed were (1) eliminated breeding to a boar line that had poor leg conformation, (2) reduced or eliminated ractopamine use, and (3) started a program that required producers to walk through the finishing pens every day. This trained the pigs to quietly get up and walk away from the person walking amongst them. These observations clearly showed how on-farm factors can have detrimental effects on both handling practices and animal welfare. 

There has been much discussion about designing and building better vehicles to reduce stress on pigs during loading and transport. In Europe, many new vehicles have power lifts and movable decks to eliminate ramps for loading and unloading pigs. In the eastern U.S., the height of almost all trucks is restricted to 13 feet 6 in. (4.11 m) due to low bridges [50]. The vehicle will be too tall if two decks of cattle are positioned above the wheels on a level floor. Two decks of cattle are accommodated in a compartment between the axles [51]. This design has internal ramps to load and unload the animals on and off the two decks. These trailers work well for cattle, but some pigs have difficulty negotiating the ramps. This has resulted in a straight trailer design for pigs that have two decks located over the top of the wheels. Use of these trailers is limited to pigs, sheep, or other small animals. Many independent truckers in the U.S. who own their own vehicles need to have the flexibility of transporting both cattle and pigs in the same trailer. These owner–operators will usually use a cattle trailer that has internal ramps to transport pigs. Some people who are concerned about animal welfare believe that this trailer design should not be used for pigs.

In the spring of 2021, the author visited an abattoir that processed pigs that lived outdoors. All of the pigs arrived in cattle trailers that had the internal ramps. The author watched four trailers unload at this abattoir. The pigs moved easily up the ramp from the belly compartment and down the ramp from the top compartment. There was zero use of electric prods and none of the animals fell down during unloading. For these pigs, the cattle trailers with the internal ramps were satisfactory. People who are concerned about animal welfare or meat quality need to think about how to improve handling. The question is: Do you improve the pig, or should you improve the design of the vehicle? This recent experience reinforced my opinion that many of the intensively raised pigs have become very difficult to handle. It is the author’s opinion that the pig needs to be improved by changing breeding, feeding, and production practices. The author is not suggesting raising all finishing pigs outside. What is being suggested is that the emphasis needs to be on improving the pigs, so they are stronger and more willing to move. One study showed that exercising pigs improved ease of handling [52]. Possible ways of doing this are genetic selection and regular moving of pigs on the farm. During the springtime observation of many pigs in the stockyard (lairage), there was only one group that had poor leg conformation. The author warned the company that they need to keep working with producers to breed pigs that have good leg conformation.

## 6. Conclusions

Previous research clearly shows that to preserve meat quality and maintain good animal welfare, cattle and pigs should move easily with a minimum use of aversive driving methods such as electric prods, tail twisting, or hitting. The meat industry needs to address increasing problems with on-farm factors that may make cattle and pigs more difficult to move at the abattoir. Lame animals that have difficulty walking are more difficult to handle. Some of the on-farm factors that may contribute to these problems are poor leg conformation in both pigs and cattle, housing finishing cattle for long periods on concrete, poor management of dairy cow cubicles, or feeding high doses of beta-agonists. To improve both animal welfare and meat quality, producers need to correct these problems.

## Data Availability

There is no data associated with this paper.

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
