# Peer review of "Cattle and Pigs Are Easy to Move and Handle Will Have Less Preslaughter Stress"

_foods, 2021, doi:10.3390/foods10112583_

Round 1

Reviewer 1 Report

The author has covered a subject that is likely to attract a wider readership, I have some minor suggestions/revisions.

  1. Under section 2.1, insert 'to' between 'reluctant' and 'move'
  2. The author has highlighted the distribution of leg conformation charts to avoid selecting animals with poor leg structural conformation, but it may useful if the author suggests other interventions for preventing this problem
  3. The author has mention of shortening the period of time animals spend on concrete floor to reduce lameness, can the author cite published literature to support their personal observations. Is there a guide to the length of period (say days or weeks) that can lead to a reduction in lameness, this will be useful to farmers and other readers of the paper.
  4. There is no clear conclusion on the use of growth promotants. There is suggestion that even using the recommended dose will lead to welfare issues. Is avoidance the best solution? The author needs to make this clear

Author Response

Reviewer 1 had four items that needed to be address.  The first one was a single grammar correction and it has been corrected.

To address the second concern, two additional causes of lameness in dairy cows were added.  They are poor body condition and slippery floors.

The third concern was adding published literature on guidelines for the length of time that cattle can be housed on concrete to reduce lameness.  These guidelines do not exist and a clear statement about the need for more research has been added.

To address the fourth concern about growth promotants, additional information on beta-agonists has been added. Beta-agonists are banned in some countries and legal in others  It is important for researchers in countries when they are legal to be aware of possible welfare problems.

Reviewer 2 Report

The review is clear, very well focused, providing a comprehensive and up to date state-of-the-art. At the same time, the manuscript gives practical issues that the modern slaughter plants have been facing in the last decades, suggesting practical improvements in abattoir facilities and modification of on-farm housing and feeding practices.

Author Response

Review Dated September 18, 2021 – Reviewer 2

The title was changed to Cattle and Pigs That are Easy to Move and Handle Will Have Less Preslaughter Stress.  The work livestock in the title was replaced with cattle and pigs to address Reviewer 2’s concern that my commentary did not cover all types of livestock.

To address Reviewer 2’s concerns that the detrimental effects of electric prods have already been extensively reviewed – additional statements have been added to the Abstract, Introduction, and Conclusions. The statements make it clear that previous research has already shown that electric prods lead to higher stress. 

To address Reviewer 2’s concerns about critical analysis, the conclusions have been revised.  The conclusions now outline some of the major on-farm factors that may contribute to making cattle and pigs more difficult to move and handle.

Reviewer 3 Report

It seems that the authors intended to write a “mini-review” about the pre-slaughter handling of livestock.

Some concerns:

  • Mostly swine and some cattle were considered.
  • The use of electric prods has not been considered for ages in the industry.
  • The fact that the use of electric prods leads to higher stress levels has been already widely described and reviewed in the literature.
  • The stress levels and their implication on meat quality have been also widely reviewed in the literature.

The manuscript lacks novelty. It must be improved with critical analyses and comments.

Author Response

Review Dated September 19, 2021 – Reviewer 3

Reviewer 3 had four items that needed to be address.  The first one was a single grammar correction and it has been corrected.

To address the second concern, two additional causes of lameness in dairy cows were added.  They are poor body condition and slippery floors.

The third concern was adding published literature on guidelines for the length of time that cattle can be housed on concrete to reduce lameness.  These guidelines do not exist and a clear statement about the need for more research has been added.

To address the fourth concern about growth promotants, additional information on beta-agonists has been added. Beta-agonists are banned in some countries and legal in others  It is important for researchers in countries when they are legal to be aware of possible welfare problems.

Round 2

Reviewer 3 Report

It seems that the authors intended to write a “mini-review” about the pre-slaughter handling of livestock.

Some concerns:

  • Mostly swine and some cattle have been considered.
  • The use of electric prods has not been considered for ages in the industry.
  • The fact that the use of electric prods leads to higher stress levels has been already widely described and reviewed in the literature.
  • The stress levels and their implication on meat quality have been also widely reviewed in the literature.

The manuscript lacks novelty. It must be improved with critical analyses and comments.

Author Response

The title was changed to Cattle and Pigs That are Easy to Move and Handle Will Have Less Preslaughter Stress.  The work livestock in the title was replaced with cattle and pigs to address Reviewer 2’s concern that my commentary did not cover all types of livestock.

To address Reviewer 3’s concerns that the detrimental effects of electric prods have already been extensively reviewed – additional statements have been added to the Abstract, Introduction, and Conclusions. The statements make it clear that previous research has already shown that electric prods lead to higher stress. 

To address Reviewer 2’s concerns about critical analysis, the conclusions have been revised.  The conclusions now outline some of the major on-farm factors that may contribute to making cattle and pigs more difficult to move and handle.